# IL-6 is one of the key factors in the formation of gut tissue resident memory T cells from Naïve T cells

Han G. Kim[1‡], Amanda Chan[1‡], Sinmanus Vimopatranon[2‡], Alexandre Girard[1], Andrew Jiang[1], Samuel Wertz[1], Il-Young Hwang[1], John H. Kehrl[1], Hana Schmeisser[1], Madelyn M. Seemiller[1], Paolo Lusso[1], Dawei Huang[3], Danlan Wei[1], Livia R. Goes[4], Marcelo Soares[4], Elena Martinelli[5], James Arthos[1‡], Claudia Cicala[1‡*]

1 Laboratory of Immunoregulation, National Institute of Allergy and Infectious Diseases, Bethesda, Maryland, United States of America, 2 Department of Retrovirology, Walter Reed Army Institute of Research-Armed Forces Research Institute of Medical Sciences, Bangkok, Thailand, 3 National Cancer Institute, Bethesda, Maryland, United States of America, 4 Instituto Nacional de Cancer, Rio de Janeiro, Brazil, 5 Division of Infectious Diseases, Northwestern Feinberg School of Medicine, Chicago, Illinois, United States of America

‡ HGK, AC and SV These authors contributed equally to this work. JA and CC These senior authors contributed equally on this work.
* ccicala@niaid.nih.gov

## Abstract

Tissue resident memory CD4+ T cells (T_RMs) populate mucosal sites and play a critical role in local immune responses. Gut T_RM cells persist for extended periods in the gut mucosa where they rapidly respond to invading pathogens and provide long lasting protection. This study investigates the factors that mediate differentiation of naïve CD4+ T cells into cells presenting a gut T_RM phenotype. Naïve CD4+ T cells were cultured under conditions that mimicked mucosal environments. This included signaling through MAdCAM-1 in the presence of Retinoic Acid (RA) and TGF-β. This combination of stimuli primed naïve CD4+ T cells to adopt a T_RM phenotype. However, to fully differentiate into T_RMs an additional soluble factor provided by memory T cells was required. Our results identified IL-6 as one of the key factors that induces the expression of T_RM -associated markers, including CD69, CD103 and CCR5. This unique combination of stimuli promoted T_RM differentiation despite low level proliferation. T_RM differentiation was mediated through JAK/STAT signaling, and antagonists that target JAK/STAT pathways suppressed MAdCAM-1 mediated T_RM cell formation. Our findings revealed that MAdCAM-1 works together with TGF-β, RA and IL-6 in this process. Such information may aid in the design of next generation adjuvants and effective mucosal vaccines. Additionally, each of these factors may be targeted to treat excessive gut inflammation associated with conditions like inflammatory bowel disease. Overall, these findings provide new strategies aimed at modulating immune responses to invading pathogens and identify therapeutic approaches toward regulating gut inflammation.

**Data availability statement:** All relevant data are within the manuscript and its Supporting information files.

**Funding:** This work was supported by Intramural Research Program of National Institute of Allergy and Infectious Diseases.(to CC, JA, HK, AC, AG, SV, AJ, SW, IH, MMS, HS, DW, JK). A grant from Carlos Chagas Filho Rio de Janeiro State Science Foundation (FAPERJ – E-40/ 200.584/2022) (to LRG). The funders had no role in study design, data collection and analysis, decision to publish, or preparation of the manuscript.

**Competing interests:** The authors have declared that no competing interests exist.

## Author summary

Immunologists and vaccinologists have long been interested in strategies aimed at boosting immune responses in mucosal tissues where pathogens are first encountered. Gut Tissue resident memory T cells ($T_{RM}$s) play a central role in gut homeostasis and immunity. They provide a rapid long-lasting protection against pathogens. $T_{RM}$s must target harmful pathogens and at the same time tolerate harmless commensal bacteria. This balance between immunity and tolerance in the complex environment of gut tissues is essential to the maintenance of gut homeostasis. In this study we have identified key factors present in the gut tissue milieu that are involved in $T_{RM}$ differentiation from naïve T cells. These findings point to new therapeutic approaches that can potentially target immune responses in gut tissues and may help in developing effective mucosal vaccines. Conversely, these factors may be targeted in excessive gut inflammation involved in conditions like inflammatory bowel disease (IBD).

## Introduction

Tissue resident memory T ($T_{RM}$) cells [1,2] reside in tissues, including skin, lung and gut mucosa, for extended periods of time (up to years) [2–6]. These cells exhibit a distinct transcriptional program [7,8] and serve as sentinels that initiate rapid secondary immune responses against invading pathogens. CD8+ $T_{RM}$ cells have been extensively described, while CD4+ $T_{RM}$ cells remain less well characterized. CD103 and CD69 are the canonical cell surface markers used to identify $T_{RM}$ cells. CD103 is the α chain of $\alpha_E\beta_7$ which functions as a mucosal retention receptor through binding to E-cadherin [9]. The persistent expression of CD69 on $T_{RM}$ cells antagonizes sphingosine-1-phosphate receptor-1 (S1PR1), a receptor that promotes lymphocyte migration out of tissues [10–12].

Two soluble factors, transforming growth factor β (TGF-β) and retinoic acid (RA) play key roles in $T_{RM}$ differentiation. TGF-β is a pleiotropic cytokine that suppresses lymphocyte activation. It also upregulates CD103 expression [13,14]. RA is a vitamin A metabolite that is produced by dendritic cells but may also be produced by intestinal epithelial cells [15]. RA drives the expression of integrin $\alpha_4\beta_7$ ($\alpha_4\beta_7$) and CCR9, both of which mediate lymphocyte homing to gut tissues [16]. Of note, RA works synergistically with TGF-β to generate intestinal $T_{RM}$ cells [17,18].

Constant exposure to dietary and bacterial antigens presents a unique challenge to the gut immune system, which requires a finely balanced control between inflammation and tolerance [19,20]. One way in which gut lymphocytes regulate immune responses involves utilizing an array of costimulatory receptors that initiate distinct cellular programs. A key costimulatory interaction involves mucosal vascular addressin cell adhesion molecule 1 (MAdCAM-1) [21–24] binding to $\alpha_4\beta_7$, an integrin receptor expressed on various lymphocyte subsets. $\alpha_4\beta_7$ is expressed uniformly on naïve CD4+ T cells, but variably on memory CD4+ T cells subsets. The high endothelial venules of Peyer's patches, mesenteric lymph nodes, and intestinal lamina propria express

MAdCAM-1, which is upregulated in response to inflammation [25,26]. It is this localized expression of MAdCAM-1 that underlies the assignment of $\alpha_4\beta_7$ as the principal lymphocyte gut homing receptor. Both RA and interleukin-7 induce upregulation of $\alpha_4\beta_7$, thus conferring upon T cell's gut-homing properties [16,27]. In addition to its trafficking function, MAdCAM-1 delivers costimulatory signals to T cells through $\alpha_4\beta_7$, which promotes cellular proliferation [23,28].

IL-6 has a dual role in GI inflammation. While it can contribute to chronic inflammation in IBD, it also plays a role in maintaining gut health and promoting tissue repair. By regulating the balance between pro- and anti-inflammatory responses in T cells, IL-6 regulates chronic GI inflammatory environment and these effects depend on the target cells [29,30]. Understanding the specific context and timing of IL-6's actions is crucial for developing effective therapies for GI disorders.

Variations in local tissue-milieu cause $T_{RM}$ cells to differentiate into diverse $T_{RM}$ cell subsets [4,6,31]. Along with different costimulatory signals, tissue specific soluble factors (e.g., cytokines, chemokines, metabolites) determine the differentiation pathways that $T_{RM}$ cells follow. How various tissue-specific factors influence this process is a subject of active investigation. Initial reports indicated that CD8+ $T_{RM}$ cells derive from effector memory CD8+ T cells. While more recent reports suggest that they can also emerge directly from naïve CD8+ T cells, significant gaps remain in our understanding of their ontogeny [4,6,32]. In this report, we describe a distinctive CD4+ $T_{RM}$ cells differentiation pathway originating from naïve CD4+ T cells. Differentiation through this pathway is mediated by MAdCAM-1 and involves an apparent synergy between a number of soluble factors including IL-6, RA and TGF-β. These findings hold the potential to inform new therapeutic approaches to the treatment of diseases involving gut immunity.

## Results

### Memory CD4+ T cell soluble factors promote the differentiation of MAdCAM-1 costimulated CD4+ naïve T cells into $T_{RM}$s

In a previous study we demonstrated that costimulating primary CD4+ T cells with MAdCAM-1 (combined with plate immobilized anti CD3 mAb) in the presence of RA for 7 days with the addition of TGF-β for the last 3 days, generates cells with a CD69+/CD103+ $T_{RM}$ phenotype [24]. In these experiments we noted that in cultures containing only purified CD45RO- cells (naïve), MAdCAM-1 costimulation inefficiently generated CD69+/CD103+ $T_{RM}$ cells relative to cultures that included both CD45RO+ and naïve cell subsets (S1 Fig). CD45RO expression is commonly used to define memory T cells. Although not perfect in all circumstances, CD45RO is largely consistent with memory and sufficient in this study. We subsequently examined how variation in relative frequencies of naïve and memory CD4+ T cell subsets among donor PBMCs influenced the efficiency of $T_{RM}$ cell generation. We purified bulk CD4+ T cells from 43 independent donors and costimulated them with MAdCAM-1 in the presence of RA and TGF-β as described above. For each of these donors we plotted the starting frequency, before culture, of naïve CD4+ T cells at day 0 on the x-axis and the frequency of CD69+/CD103+ $T_{RM}$ cells at day 7 on the y-axis. This analysis revealed a weak positive correlation ($R^2$ = 0.053) (Fig 1A), suggesting that the naïve/memory ratio may affect $T_{RM}$ differentiation. We confirmed this relationship by combining different ratios of purified naïve T cells and purified memory T cells from the same donor (Fig 1B). As the ratio of naïve to memory cells on day 0 increased, the frequency of $T_{RM}$ cells on day 7 also increased. However, when we removed memory cells and employed purified naïve cell cultures, we observed a sharp reduction in the frequency of $T_{RM}$ cells (Fig 1B). We repeated this experiment with purified naïve cultures from 8 donors which yielded an average of 9.40 ± 8.35 CD69+/CD103+ cells (median 5.98 ± 12.3) (Fig 1C). In contrast, costimulation of matched bulk cultures (n = 8) yield an average of 25.3 ± 12.2 double positive cells (median 25.8 ± 23.9). These findings suggest that MAdCAM-1 costimulation can generate $T_{RM}$ cells from naïve CD4+ T cells, but a small number of memory T cells facilitates this process.

We next asked how the relative frequencies of naïve and memory CD4+ T cells impacted $T_{RM}$ differentiation in the context of other costimulatory signals. Cells were costimulated with VCAM-1, which binds both $\alpha_4\beta_1$ and $\alpha_4\beta_7$, and CD28 mAb. For bulk CD4+ T cell cultures (n = 43), VCAM-1 and CD28 mAb induced similar frequencies of $T_{RM}$ cells compared

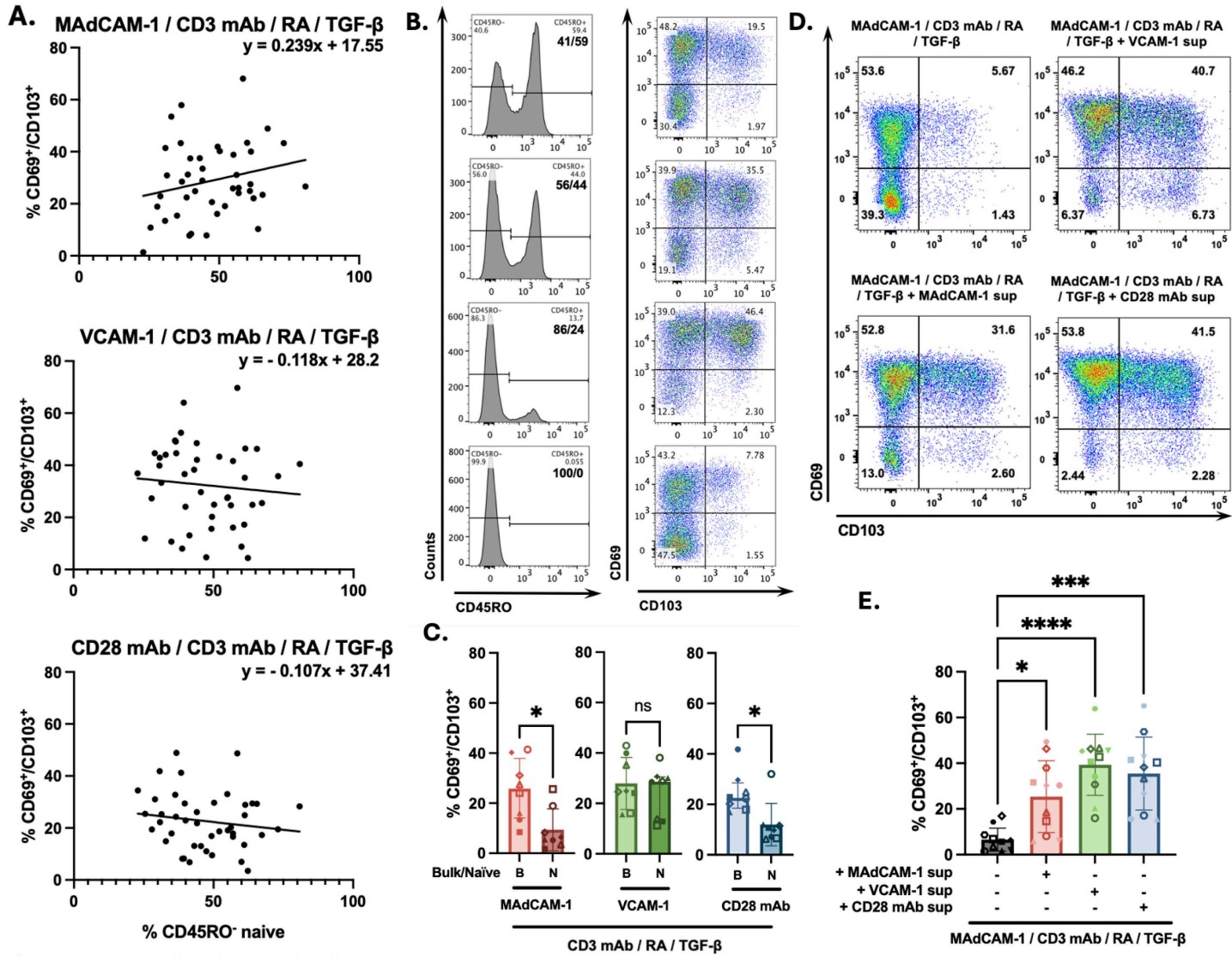

**Fig 1. Soluble factors promote MAdCAM-1 costimulated naïve CD4 $^+$ T cells to express T$_{RM}$ cell surface markers.(A)** The frequency (%) of CD45RO$^-$ CD4$^+$ T cells on day 0 (x-axis) and the frequency of CD69$^+$/CD103$^+$ cells on day 7 in multiple independent donors (y-axis) following costimulation of bulk CD4$^+$ T cells with MAdCAM-1 (upper), VCAM-1 (middle) or CD28 mAb (lower) in the presence of RA and TGF-β (closed circles, n=43). The solid black line and equation represent a linear regression of that includes only bulk cultures. The dashed line represents a quadratic polynomial regression that includes both bulk and purified CD45RO$^-$ cells. Linear correlation coefficients are provided for each panel. **(B)** CD69/CD103 coexpression from a representative donor following MAdCAM-1 costimulation of CD45RO$^-$/CD45RO$^+$ CD4$^+$ T cell cultures reconstituted at different ratios. Left panels indicate the approximate ratio of CD45RO$^-$ to CD45RO$^+$ cells on day 0. Right panels indicate the coexpression of CD69 (y-axis) and CD103 (x-axis) on day 7, with the frequency of double positive cells indicated. **(C)** Frequency of CD69$^+$/CD103$^+$ cells following MAdCAM-1, VCAM-1, and CD28 mAb costimulation of matched bulk (B) and naïve (N) CD4$^+$ T cell cultures (n=8) harvested at day 7. Y-axis indicates % double positive cells. **(D)** Coexpression of CD69 and CD103 following MAdCAM-1 costimulation of purified CD45RO$^-$ cells in the absence (UL panel) or presence of VCAM-1 (UR), MAdCAM-1 (LL) or CD28 mAb (LR) bulk culture supernatants, from a representative donor. **(E)** Treatment of purified CD45RO$^-$ cells as in panel D (n=11) harvested at day 7. Error bars indicate standard deviation (*: p<0.05, **: p<0.01, ***: p<0.001, ****: p<0.0001).

to MAdCAM-1 (Fig 1C). Unlike MAdCAM-1, a positive correlation was not observed in VCAM-1 and CD28 mAb treated cells (R$^2$=0.009 and 0.022, respectively) (Fig 1A). VCAM-1 and CD28 mAb also exhibited stronger costimulatory signaling with greater T cell viability compared to MAdCAM-1 cultures (n=10) (average MAdCAM-1=29.15% ± 10.45,

VCAM-1 = 50.72% ± 11.18, CD28 mAb = 52.45% ± 12.97) (S2 Fig). When comparing bulk cultures to purified naïve CD4$^+$ T cell cultures from the same donors (n = 8), VCAM-1 exhibited a greater capacity to generate T$_{RM}$ cells (average naïve: 24.3 ± 10.1 vs bulk 27.9 ± 10.3) while CD28 mAb -mediated T$_{RM}$ cells formation was reduced, (average naïve:12.0 ± 8.39 vs bulk: 24.6 ± 8.05) (Fig 1C). To determine if memory T cells provided a soluble factor(s), we added supernatants from day 4 stimulated bulk CD4$^+$ T cells (stimulated with either MAdCAM-1, VCAM-1, or CD28 mAb) to MAdCAM-1 costimulated naïve CD4$^+$ T cell cultures. All 3 bulk supernatants provided a soluble factor(s) that significantly "rescued" T$_{RM}$ cell formation from naïve cell cultures (Fig 1D, 1E). We noted that CD28 mAb and VCAM-1 supernatants were more efficient (Fig 1E). We concluded that MAdCAM-1 provides a costimulatory signal that prompts naïve CD4$^+$ T cells to differentiate into T$_{RM}$ cells, and is facilitated by a factor(s) provided by memory CD4$^+$ T cells.

### MAdCAM-1 costimulation in the presence of TGF-β promotes T$_{RM}$ cell formation despite low level proliferation

We next investigated to what extent MAdCAM-1 -mediated T$_{RM}$ differentiation of naïve cells depended upon the proliferation of either subset of CD4$^+$ T cells. While α$_4$β$_7$, the MAdCAM-1 receptor, is uniformly expressed on naïve CD4$^+$ T cells, it appears on only a small subset of memory CD4$^+$ T cells [33,34]. We labelled purified naïve and memory CD4$^+$ T cells with CFSE and Far Red respectively and recombined the two cell types at a ratio favorable for T$_{RM}$ cell formation (2:1, naïve to memory). Because TGF-β exerts an antiproliferative effect on lymphocytes [13,35,36], we cultured the cells in the absence or presence of exogenous TGF-β. Cultures were stimulated as described above and analyzed for dye dilution proliferation on day 7 using flow cytometry. Proliferation was detected in both naïve and memory cell subsets, with or without TGF-β (Fig 2A). We obtained average division indexes for naïve (Fig 2B) and memory (Fig 2C) cell subsets from 4 different donors. MAdCAM-1 promoted proliferation to a greater degree in naïve cells (naïve: 0.65 ± 0.2 vs memory: 0.24 ± 0.1), while TGF-β significantly suppressed this proliferation in both subsets (Fig 2B, 2C). For comparison we stimulated cells with VCAM-1 and CD28 mAb. Proliferation was induced in both naïve and memory subsets, and was not suppressed by the addition of TGF-β. To better understand how proliferation was linked to T$_{RM}$ differentiation we plotted the percent expression of CD103 and CD69 against successive cell divisions. MAdCAM-1 costimulation, in the absence of TGF-β, induced few cells (< 5%) to express CD103 (Fig 2D). The addition of TGF-β significantly upregulated CD103 expression in cells undergoing one or more division cycles (Figs 2D and S3A). CD69 expression was somewhat reduced by TGF-β (~17.0%), but cells remained CD69$^+$ throughout their successive divisions (Figs 2D and S3B). MAdCAM-1 in the presence of RA and TGF-β uniquely upregulated the expression of CCR5 (Figs 2D and S3C). This effect on CCR5 expression was not observed with VCAM-1 or CD28 mAb (Figs 2D and S3C). We concluded that the combination of MAdCAM-1, RA and TGF-β was distinct in its capacity to upregulate the expression of two key gut T$_{RM}$ markers (CD103 and CCR5) on naïve CD4$^+$ T cells. This occurred despite low level cell proliferation.

### IL-6 promoted MAdCAM-1 mediated T$_{RM}$ cells differentiation of naïve CD4$^+$ T cells

As described above, MAdCAM-1 costimulation through α$_4$β$_7$ induces naïve CD4$^+$ T cells to differentiate into T$_{RM}$s. Additionally, unidentified soluble factors present in bulk culture supernatants promote the expression of canonical T$_{RM}$ surface markers (Fig 1E). As candidate soluble factors we tested 3 proinflammatory cytokines: TNF-a, IFN-γ, and IL-6, and an anti-inflammatory cytokine, IL-10. We purified naïve CD4$^+$ T cells from 6 independent donors and stimulated them as described in Fig 1E except that we added recombinant cytokines on day 4 in place of culture supernatants. In the absence of any added factor, naïve cells stimulated with MAdCAM-1, RA and TGF-β poorly generated T$_{RM}$ cells (Fig 3A). While we did not rescue T$_{RM}$ cell generation with the addition of TNF-a, IFN-γ or IL-10 (Fig 3A), when we added IL-6, naïve cells differentiated into T$_{RM}$ cells at a level comparable to those observed when adding bulk cultures supernatants (Fig 3A). CD4$^+$ CD45RA$^+$ naïve T cells also express higher levels of both IL6 receptor (IL-6Rα) and the signal transducer gp130 (IL-6Rβ) when compared to CD45RO$^+$ memory T cells (S4 Fig). To further define the role of IL-6, we tested whether the IL-6 receptor antagonist tocilizumab [37], inhibited T$_{RM}$ cell formation. Purified naïve cells were costimulated with MAdCAM-1,

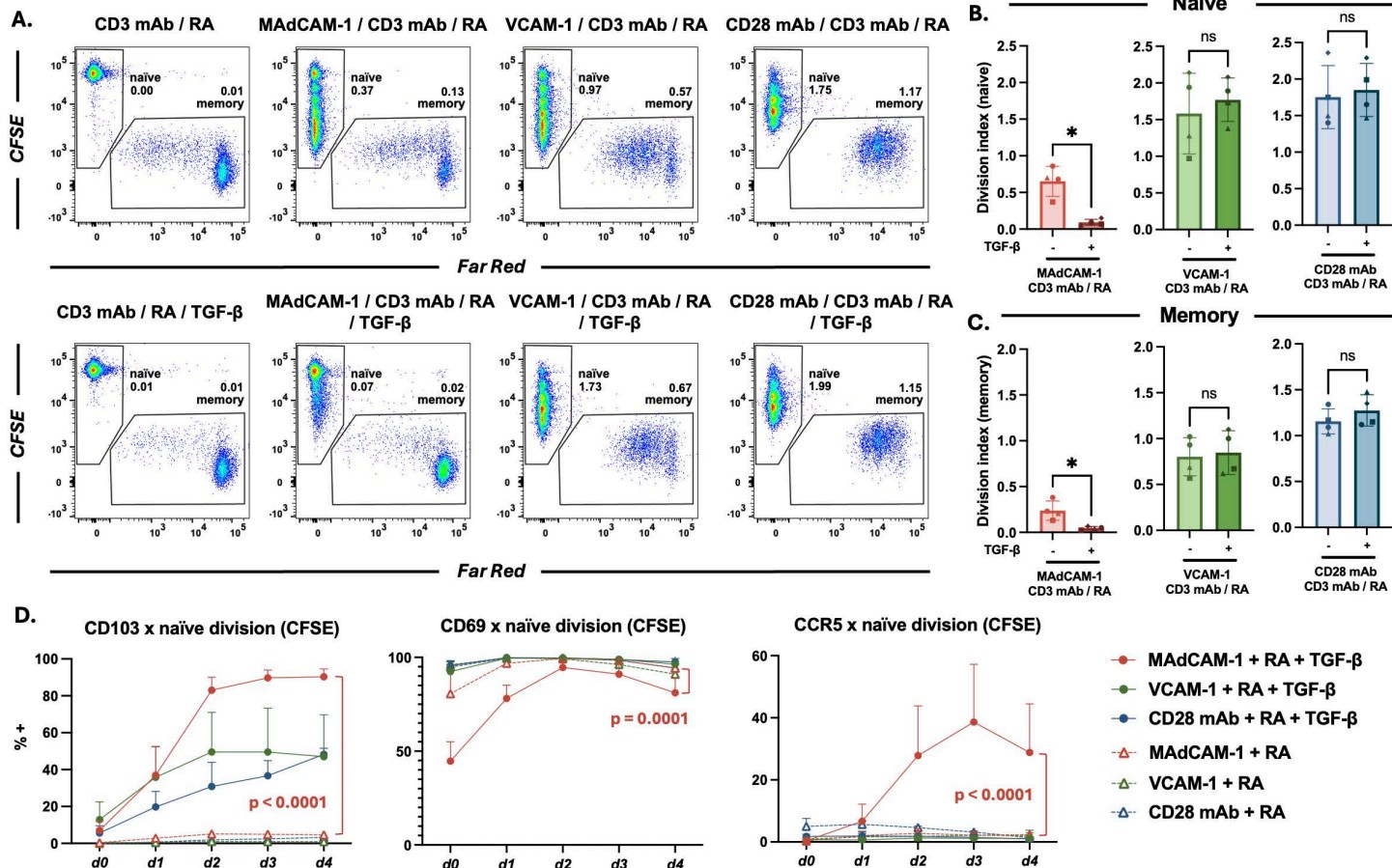

**Fig 2. Naïve and memory CD4 + T cell proliferation in response to MAdCAM-1, VCAM-1, and CD28 mAb costimulation. (A)** Flow cytometric dot plots of CD4+ T cell proliferation by dye dilution, from a representative donor, costimulated with MAdCAM-1, VCAM-1 or CD28 mAb without (upper) and with (lower) TGF-β. Reconstituted cultures included naïve cells (CD45RO-) labelled with CFSE (y-axis), and memory cells labelled with Far Red (x-axis). Division index (DI) for each population is indicated. **(B and C)** DI of naïve and memory cells from 4 donors (y-axis), costimulated as in panel A with and without TGF-β, as indicated (*: p<0.05). **(D)** CD4+ T cells costimulated as in panel A. Average frequency of CD103, CD69 and CCR5 expression on naïve cells in each division cycle, in the presence or absence of TGF-β as indicated (n=3). *d0* represents non-divided cell populations with each subsequent *dn+1* representing an additional division. Comparisons between MAdCAM-1+RA and MAdCAM-1+RA+TGF-β are provided. Error bars indicate standard deviation (p<0.0001).

RA and TGF-β, and bulk culture supernatants were added on day 4 and tocilizumab was added on both day 0 and day 4. Tocilizumab addition resulted in a significant reduction in the frequency of CD69+/CD103+ cells in all donors tested (mean percent reduction=23.34±10.89) (Fig 3B). The failure to completely inhibit the supernatant activity suggested that additional factors present in the bulk supernatants also promoted T$_{RM}$ cell formation (Fig 3B).

Next we determined whether IL-6 impacted T$_{RM}$ cell generation following VCAM-1 and CD28 mAb costimulation.

Naïve cells were costimulated with each ligand in the presence or absence of exogenous IL-6 and the frequency of CD69+/CD103+ cells was measured on day 7. IL-6 promoted VCAM-1 mediated upregulation of CD69+/CD103+ cells (Fig 3C), but it had no significant effect on CD28 mAb mediated generation of T$_{RM}$ cells (Fig 3C). Thus, IL-6 promoted T$_{RM}$ differentiation in response to the two costimulatory ligands that signal through integrins containing the α$_4$ chain, while having no significant effect on CD28 mAb -mediated induction of T$_{RM}$ cells. To determine whether adding IL-6 earlier in the culture would further enhance T$_{RM}$ differentiation we modified our protocol and added IL-6 on day 0 and again on day 4. This

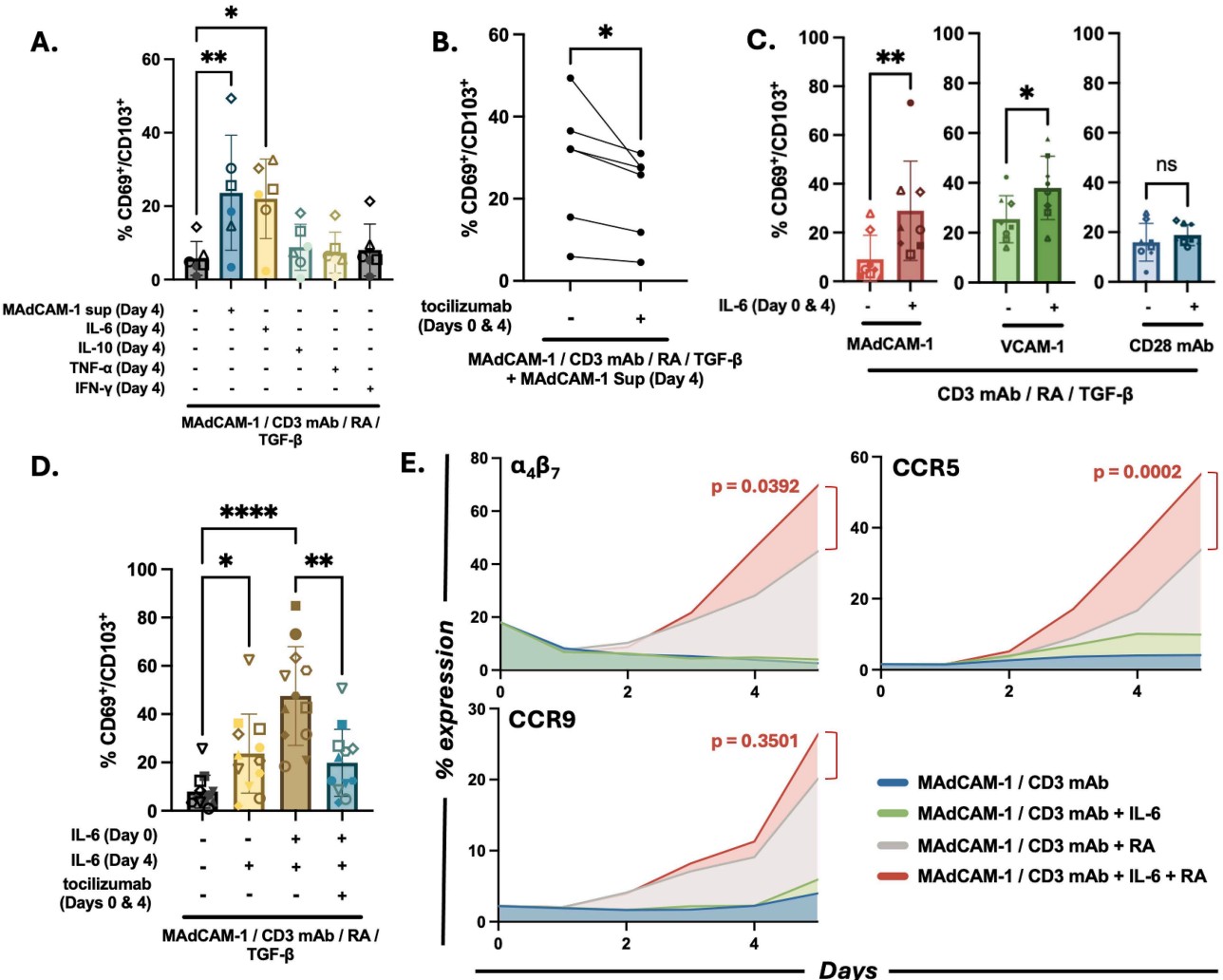

**Fig 3. IL-6 promotes MAdCAM-1 mediated T_RM differentiation of naïve CD4 + T cells.(A)** Flow cytometric analysis comparing CD69+/CD103+ upregulation on naïve CD4+ T cells costimulated with MAdCAM-1, RA and TGF-β in the absence vs presence of exogenous IL-6, IL-10, TNF-α, and IFN-γ, as indicated. Treatment with day 4 culture supernatant included for comparison (n = 6). **(B)** Coexpression of CD69 and CD103 following MAdCAM-1 costimulation (with RA and TGF-β) of bulk CD4+ T cells in the absence or presence of tocilizumab (added on day 0 and day 4) as indicated (n = 6). **(C)** Coexpression of CD69 and CD103 following VCAM-1 or CD28 mAb costimulation in the absence or presence of exogenous IL-6 (added on day 0 and day 4). MAdCAM-1 costimulation included for comparison (n = 8). **(D)** Average CD69/CD103 coexpression on MAdCAM-1 costimulated cells treated with IL-6 on day 4 vs day 0 and day 4 (n = 12). Tocilizumab addition included as a control. **(E)** Flow cytometric time course (days 0-5) (x-axis) of α_4β_7, CCR9 and CCR5 expression on purified naïve CD4+ T cells costimulated with MAdCAM-1 in the absence or presence of IL-6, RA, or IL-6 + RA. Average % positive cells (n = 3) is reported (y-axis). Error bars indicate standard deviation (*: p < 0.05, **: p < 0.01, ***: p < 0.001, ****: p < 0.0001).

modification increased the frequency of T_RM cells in 12 independent donors compared to adding IL-6 only on day 4 (Fig 3D). Addition of tocilizumab on day 0 significantly suppressed the generation of T_RM cells. These observations suggested that IL-6 priming of naïve CD4+ T cells began during the early stages of T_RM differentiation.

Finally, we asked how IL-6, in the context of MAdCAM-1 costimulation, influences the expression of three gut associated receptors (α_4β_7, CCR5 and CCR9). Naïve CD4+ T cells from 3 independent donors were costimulated with MAdCAM-1 alone, MAdCAM-1 + IL-6, MAdCAM-1 + RA, and MAdCAM-1 + IL-6 + RA and surface receptor expression was

measured over time by flow-cytometry. IL-6 combined with MAdCAM-1 (without TGF-β and RA) had little effect on the expression of these markers (Fig 3E). However, the combination of IL-6 and RA induced $\alpha_4\beta_7$ and CCR5 expression to a greater extent than either factor alone (CCR5 p = 0.0002; $\alpha_4\beta_7$ p = 0.0392), suggesting a synergistic effect between these two soluble factors (Figs 3E, S5). In contrast, CCR9 expression was enhanced primarily by RA, whilst IL-6 addition had little impact. Overall, these findings indicated that, in the context of MAdCAM-1 costimulation in combination with RA and TGF-β, IL-6 induces naïve CD4$^+$ T cells to adopt a $T_{RM}$ phenotype. Additionally IL-6 appears to work cooperatively with RA in promoting the expression of gut tissue receptors.

**MAdCAM-1 bulk supernatants prime naïve CD4$^+$ T cells to differentiate into $T_{RM}$ cells through JAK/STAT signaling**

Many cytokines including IL-6 signal by activating the JAK/STAT pathway. IL-6 signaling specifically triggers STAT1 and STAT3 phosphorylation [38] (Fig 4A). To address the involvement of JAK/STAT signal transduction pathway in the generation of $T_{RM}$ cells, we briefly costimulated naïve cells with MAdCAM-1 and RA in the absence or presence of bulk culture supernatants. We prepared lysates that were then analyzed by Western blot using a panel of anti phospho STAT mAbs. Cells treated with IL-6 were tested as a control. The bulk supernatants rapidly (~10') induced STAT1, STAT2 and STAT3 phosphorylation (Fig 4B). We next analyzed the same lysates with total STAT1 and STAT3 mAbs along with an actin mAb, and determined that the overall levels of STAT1 and STAT3 did not change during the time course. In contrast and as expected, IL-6 induced STAT1 and STAT3 phosphorylation [39] but failed to induce STAT2 phosphorylation. This finding is consistent with the inability of tocilizumab to fully neutralize the activity of the bulk culture supernatants and suggested the presence of additional $T_{RM}$ differentiation factors responsible for the STAT2 phosphorylation.

A good candidate was Interferon-β (IFN-β), one of the cytokines that delivers signals through STAT2 phosphorylation [40,41] and is reported to modulate CD103 expression in mice [42]. We tested whether IFN-β could facilitate the formation of $T_{RM}$ cells from naïve CD4$^+$ T cells costimulated with MAdCAM-1. Purified naïve cells from five independent donors were stimulated with MAdCAM-1 in the presence of RA and TGF-β with and without the addition of exogenous IFN-β. IL-6 stimulation was added as a control. On day 7, we analyzed the cells for CD69/CD103 coexpression by flow cytometry. Similar to IL-6, IFN-β induced the coexpression of these two canonical $T_{RM}$ markers (Fig 4C), a finding that is suggestive of STAT2 involvement. However, because IFN-β mediates phosphorylation of multiple STATs, we cannot exclude the possibility that upregulation of $T_{RM}$ markers was solely mediated by STAT1 and STAT3 phosphorylation.

To firmly establish a role of JAK/STAT signaling in $T_{RM}$ differentiation we employed two JAK phosphorylation inhibitors, ruxolitinib [43] and AZD1480 [44]. Bulk CD4$^+$ T cells from 6 or more donors were treated with MAdCAM-1 in combination with TGF-β, and analyzed for CD69/CD103 coexpression by flow cytometry, as described above. VCAM-1 and CD28 mAb were tested as controls. Ruxolitinib and AZD1480 were added on both day 0 and day 4 or only on day 4. For MAdCAM-1 costimulation both inhibitors, when added at day 0, resulted in a near complete inhibition of $T_{RM}$ cell formation (Fig 4D and 4E). The addition of ruxolitinib or AZD 1480 on day 4 only resulted in a 50.62% ± 24.07%, and 38.70% ± 23.42% inhibition respectively. For VCAM-1 and CD28 mAb costimulations, the pattern of inhibition differed. The addition on day 4 resulted in a significant increase in $T_{RM}$ cells generation, whereas addition on both day 0 and day 4 resulted in a partial inhibition. Overall, these findings suggested that MAdCAM-1 costimulation primes the cells to differentiate into $T_{RM}$ cells, while soluble factors that signal through a JAK/STAT pathway bring this process to completion. Moreover, MAdCAM-1 employs a pathway that is distinct from that used by VCAM-1 and CD28 mAb, underscoring that different pathways can lead to $T_{RM}$ cell differentiation. Overall, we found that MAdCAM-1 in combination with RA primes naïve CD4$^+$ T cells to express markers of gut $T_{RM}$ cells. The addition of IL-6 and TGF-β promotes this process even further. The implications and limitations of our findings are discussed below.

 

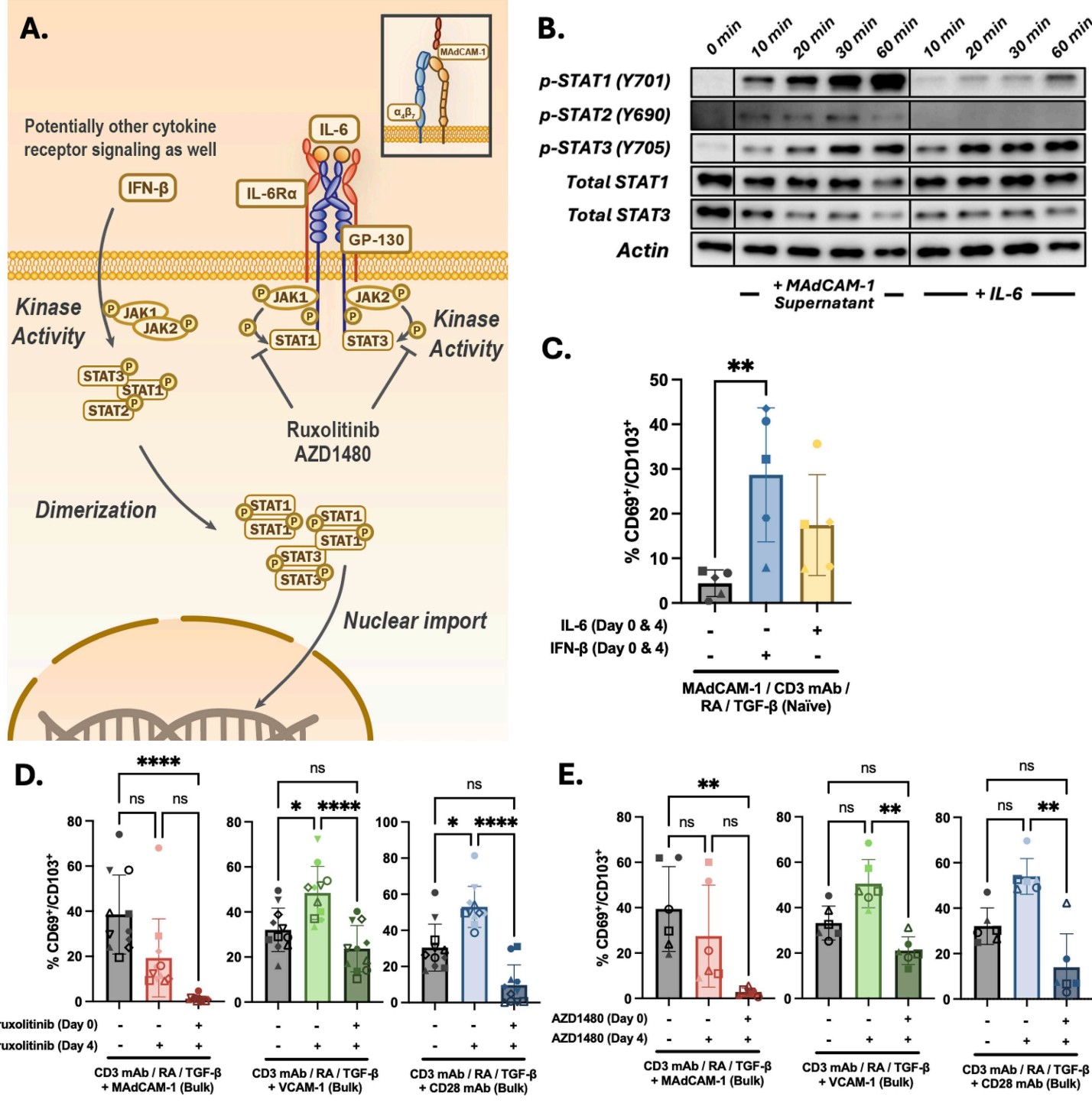

**Fig 4. JAK/STAT signaling and T$_{RM}$ CD4$^+$ T cell formation.(A)** Schematic of JAK/STAT signaling pathways. **(B)** Western blot analysis of lysates from naïve CD4$^+$ T cell cultures costimulated with MAdCAM-1 with the addition of day 4 bulk CD4$^+$ T cell supernatants (left) or exogenous IL-6 (right). Lysates harvested from 0-60 min post treatment were reacted with anti phospho -STAT1, -STAT2 and -STAT3 mAbs and total anti -STAT1 and -STAT3 mAbs. Actin mAb staining included as a control. **(C)** Flow cytometric analysis of naïve CD4$^+$ T cells (n = 5) costimulated with MAdCAM-1 (+ RA and TGF-β) in the absence or presence of either IFN-β or IL-6, as indicated. Cytokines added on both day 0 and day 4. **(D)** Flow cytometric analysis of CD69/CD103 coexpression of bulk CD4$^+$ T cells (n = 11) following MAdCAM-1, VCAM-1 or CD28 mAb costimulation (+ RA and TGF-β) in the absence or presence of ruxolitinib. Ruxolitinib added on day 0 or day 0 and 4, as indicated. **(E)** Flow cytometric analysis of CD69/CD103 coexpression of bulk CD4$^+$ T cells (n = 6)

following MAdCAM-1, VCAM-1 or CD28 mAb costimulation (+ RA and TGF-β) in the absence or presence of AZD1480. AZD1480 added on day 0 or day 0 and 4, as indicated. Error bars indicate standard deviation (*: p < 0.05, **: p < 0.01, ***: p < 0.001, ****: p < 0.0001).

## Discussion

We previously reported that MAdCAM-1, when combined with RA, provides a costimulatory signal that primes a subset of peripheral CD4$^+$ T cells to express canonical T$_{RM}$ markers (CD69 and CD103), along with α$_4$β$_7$, CCR5 and CCR9, three receptors associated with trafficking to the gut associated lymphoid tissue (GALT) [24]. The addition of TGF-β further upregulated the expression of T$_{RM}$ markers. From our initial observations it was unclear whether these cells originated from naïve or memory cell subsets (or both). Previous studies suggested that T$_{RM}$ cells originate from activated effector T cells [6,45,46]. However, recent studies have illuminated an additional pathway wherein naïve T cells can be primed to differentiate into T$_{RM}$ cells [6,32,47]. Results presented herein support this concept of priming and provide evidence that signaling through α$_4$β$_7$ plays a central role in this process. A better understanding of T$_{RM}$ cell ontogeny holds the promise to enhance our ability to develop treatments against mucosal infections and cancer, and to develop vaccines that target mucosal tissues [4,32,48].

We previously noted that the proportion of naïve and memory cells in our starting cultures impacted the efficiency with which T$_{RM}$ cells were generated. In this study we addressed the potential of both cell subsets to adopt a T$_{RM}$ phenotype. We found that bulk cultures with higher proportions of naïve cells tended to generate higher frequencies of T$_{RM}$ cells following MAdCAM-1 costimulation. However, from purified naïve cell cultures, MAdCAM-1 inefficiently generated T$_{RM}$ cells. These contrasting observations suggested that naïve cells can differentiate into T$_{RM}$ cells, but that memory CD4$^+$ T cells provided an essential soluble factor(s). We tested 4 potential factors: TNFa, IFN-γ, IL-10 and IL-6. IL-6 consistently promoted T$_{RM}$ cell formation in purified naïve cell cultures. We conclude that IL-6 is one of the soluble factors provided by memory CD4$^+$ T cells. However, incomplete inhibition of T$_{RM}$ cell formation by tocilizumab, an IL-6 antagonist, suggested the involvement of other factors. We found that IFN-β, which signals through multiple STATs including STAT2 [40,49], could also promote T$_{RM}$ cell formation. However this finding left unanswered the specific roles of different STATs in T$_{RM}$ differentiation, and the identity of additional factors remains to be determined. Nevertheless, while other factors may contribute, we are able to conclude that IL-6 helps mediate T$_{RM}$ cell formation through JAK/STAT signaling. Of note, two JAK1/2 inhibitors, ruxolitinib and AZD1480, completely inhibited T$_{RM}$ cell formation. Overall, our findings suggest that IL-6 mediates the formation of a distinct subset of T$_{RM}$ cells and that other factors may give rise to alternative T$_{RM}$ subsets.

The proportion of naïve and memory cells in starting cultures had less impact on T$_{RM}$ cell formation when those cultures were stimulated with VCAM-1 or CD28 mAb. We found a weak positive correlation between the frequency of naïve cells in starting bulk cultures and T$_{RM}$ cell formation when MAdCAM-1 was employed, but no correlation when either VCAM-1 or CD28 mAb were employed. A likely explanation for these differences involves the capacity of these two ligands to stimulate memory CD4$^+$ T cells. Their counterreceptors, α$_4$β$_1$ and CD28, are expressed on most memory CD4$^+$ T cells. In contrast, α$_4$β$_7$ is expressed on all peripheral naïve CD4$^+$ T cells but on only a small fraction of memory CD4$^+$ T cells (5%-15%) [33,34]. Consistent with this explanation, both VCAM-1 and CD28 mAb induced memory cell proliferation in a more efficient way than MAdCAM-1 (Fig 2A, 2C). While T$_{RM}$ cells were initially thought to originate mainly from effector T cells we, like others [6,32,47], have shown that naïve T cells can also give rise to CD4$^+$ T$_{RM}$ cells, highlighting the diversity of pathways that generates this T cell subset.

The naïve CD4$^+$ T cells in our starting cultures express a near uniform level of α$_4$β$_7$, typically referred to as "intermediate" expression, relative to α$_4$β$_7$ expression levels on effector memory CD4$^+$ T cells, which are typically described as α$_4$β$_7$$^{high}$ cells. As naïve cells respond to MAdCAM-1, α$_4$β$_7$ expression increases in a process that is facilitated by RA, a vitamin A metabolite that plays a central role in mucosal immune responses [16,18,50]. Ultimately, these cells upregulate α$_4$β$_7$ expression to a level comparable to that observed on effector memory CD4$^+$ T cells. They also upregulate CCR9

and CCR5, two chemokine receptors associated with gut tissue homing. As such, combining MAdCAM-1 with IL-6, RA and TGF-β promotes a phenotype that facilitates retention in gut tissues. Although MAdCAM-1 primes CD4$^+$ T cells, TGF-β drives T$_{RM}$ differentiation further. TGF-β is a pleiotropic immunosuppressive cytokine that dampens cell activation [13,35,36]. Previous studies report that TGF-β is linked to tissue residency through direct upregulation of CD103 [6,14,51,52]. Of note, RA has been reported to act in synergy with TGF-β in upregulating CD103 expression on T cells [53]. We find that TGF-β also upregulates CCR5 [22,24]; however, this only occurs in the context of MAdCAM-1 costimulation, underscoring the unique way in which MAdCAM-1 promotes gut T$_{RM}$ differentiation. Several groups report that, with respect to CD8$^+$ T cells, TGF-β and RA work synergistically in promoting tissue residency [17,18]. A similar dynamic likely holds true for CD4$^+$ T cells. Because TGF-β exerts a suppressive effect on CD4$^+$ T cells, MAdCAM-1 -mediated differentiation into T$_{RM}$ cells proceeds despite limited cell division (Figs 2 and S3). This is consistent with the unique milieu of the gut which favors a regulatory environment that controls inflammatory responses [19,20,31].

We previously reported that the CD4$^+$ T$_{RM}$ cells we generate in vitro are highly susceptible to HIV infection, due in part to the expression of $\alpha_4\beta_7$ and CCR5 [23,24]. These cells, which present a phenotype that favors retention in gut tissues, show a reduced proliferative capacity, which may facilitate the formation of latently infected cells that constitute persistent HIV-1 reservoirs. It has been established that long-lived reservoirs form in the gut within the first weeks of infection. In the acute phase of infection, $\alpha_4\beta_7^{high}$/CCR5$^+$ CD4$^+$ T cells are preferentially infected [54]. High-level viral replication in gut inductive tissue is a second feature of acute infection [55]. As such, we speculate that infected $\alpha_4\beta_7^+$ cells differentiate into T$_{RM}$ cells and subsequently establish residency in gut tissues. Of note, galunisertib, a TGF-β antagonist, reduces the size of viral reservoirs in vivo [56]. Additionally, vedolizumab, tocilizumab and JAK inhibitors may provide novel ways to prevent the formation of long-lived HIV reservoirs that reside in gut tissues [57].

Our results, as described above, indicate that factors other than IL-6 (e.g., IFN-β) can also promote T$_{RM}$ cell differentiation, and we speculate that these factors promote the differentiation of distinct T$_{RM}$ subsets. However, we have not yet identified the full complement of these factors and this represents a limitation of this study. Additionally, we assume that memory CD4$^+$ T cells are the source of IL-6, however we cannot exclude the possibility that other cells present in gut tissues provide an alternative source in vivo.

In conclusion, in this report we showed that MAdCAM-1 can prime naïve CD4$^+$ T cells to differentiate into cells presenting a gut T$_{RM}$ phenotype (S6 Fig). We further show that IL-6 can promote this process. We previously determined that vedolizumab, an $\alpha_4\beta_7$ antagonist employed as a therapeutic for inflammatory bowel diseases (IBDs), can suppress the formation of MAdCAM-1 -generated T$_{RM}$ cells [24]. Here we report that an IL-6 antagonist, tocilizumab, along with JAK inhibitors can also target T$_{RM}$ cell differentiation. These findings point to new therapeutic approaches that can potentially target immune responses in gut tissues.

## Methods and materials

### Ethics statement

All primary CD4$^+$ T cells used in this study were isolated from PBMCs collected anonymously from healthy donors from a National Institutes of Health Department of Transfusion Medicine protocol approved by the Institutional Review Board of the National Institute of Allergy and Infectious Diseases, National Institutes of Health. All study participants provided written informed consent.

### Human blood processing and CD4$^+$ T cell isolation

PBMCs were first isolated from the whole blood using lymphocyte separation medium (MP Biomedicals, Santa Clara, CA, USA). PBMCs then underwent negative isolation for both bulk CD4$^+$ T cells and purified naïve CD4 T cells (CD45RO$^-$/CD45RA$^+$)(Stem Cell Technologies, Vancouver, Canada). Following isolation, purity of CD4$^+$ isolation and percentage of CD45RO/CD45RA subsets were determined by flow cytometry.

## CD4+ T cell co-stimulation

Recombinant human MAdCAM-1-Ig and VCAM-1-Ig were purchased from R&D Systems (Minneapolis, MN, USA). Anti-CD28 mAb was purchased from Invitrogen (Carlsbad, CA, USA). Co-stimulatory ligands were biotinylated per the manufacturer's instructions using a LYNX Rapid Plus Biotin (type 2) Antibody Conjugation Kit (Bio-Rad, Hercules, CA, USA). All co-stimulation assays were carried out as previously described with the following modifications [22,23]. 96-well flat bottom cell culture-treated plates (Costa cat. # 3596) were pre-coated with 10 ng of anti-CD3 (clone OKT3) (eBioscience, San Diego, CA, USA) at 4°C for 2 hours, followed by 200 ng of neutravidin (Invitrogen, Waltham, MA, USA) at 4°C overnight in 100 µL of sterile HEPES-buffered saline (HBS). After rinsing, 200 ng of biotinylated co-stimulatory ligand was added for 1 hour at 37°C. $2 \times 10^5$ purified CD4+ T cells were then added to the coated wells in complete Roswell Park Memorial Institute (RPMI) 1640 medium with 2% L-glutamine-penicillin-streptomycin and 10% fetal bovine serum (FBS) (both from Gibco Laboratories, Gaithersburg, MD, USA) ($2 \times 10^6$ cells/mL) at 37°C, 5% carbon dioxide. Depending on the experimental condition the following soluble factors were added on either day 0 or day 4: 10nm all-trans RA (Sigma-Aldrich, St. Louis, MI, USA), TGF-β (1ng/mL), IL-6 (10ng/mL), IFN-γ (10ng/mL), IL-10 (10ng/mL), TNF-α (10ng/mL), tocilizumab (5µg/mL), ruxolitinib (1µM), AZD1480 (1µM), IFN-β (10 ng/ml). On day 4, cultures were replaced with fresh complete media. For time-course experiments, 5 wells of naïve CD4+ T cells ($2 \times 10^6$ cells/mL) were prepared per experimental condition. Cells were removed at day 1–5 with no washes during this period.

## Antibodies and flow cytometry

A catalogue of mAbs used in this study is provided in S1 Table. Antibody staining for flow cytometry was carried out in 2% FBS in 1×PBS and employed standard protocols. Data were collected on a FACS Symphony A3 (BD Biosciences, San Diego, CA, USA) and analyzed using FlowJo (BD Life Sciences).

## Proliferation by dye-dilution

Naïve and memory CD4+ T cells were negatively selected (Stem Cell Technologies, Vancouver, Canada) from PBMCs. The naïve population was labeled with 0.5µM of CellTrace CFSE while the memory population was labeled with 4µM of CellTrace Far Red. Fluorescent dyes were purchased from the CellTrace Cell Proliferation Kit (Invitrogen, Carlsbad, CA, USA). Stained cells were recombined into a ratio of [2] CFSE-Naïve to [1] Far Red-Memory, and then plated per costimulation methods listed above.

## Western blot assay

CD45RO- naïve cells were isolated and incubated with supernatants from bulk CD4+ T cells co-stimulated with MAdCAM-1 and RA for 4 days. Cell pellets were harvested at desired time points (10, 20, 30, 60 minutes) post stimulation with MAdCAM-1 and RA supernatant or IL-6, washed with PBS, and frozen. The 0 min condition received no RA, CD3 mAb, MAdCAM-1 or treatment. Cells pellets were analyzed as previously described [58]. Cell lysates were prepared using lysis buffer (20mM TRIS, 100mM NaCl, 1mM EDTA, 1% Triton X-100, 1mM DTT) with protease and phosphatase inhibitors (Thermo Scientific, Waltham, MA). 5 µL of each sample were loaded into wells of 10–20% Tris-Glycine SDS gels (Thermo Scientific, Waltham, MA) and run under reducing conditions, followed by transfer onto nitrocellulose membranes (Thermo Scientific, Waltham, MA) using the iBlot2TM system (Invitrogen, IB 1001). Antibodies used in Western blot are specified in S1 Table. Membranes were developed using SuperSignal West Femto Maximum Sensitivity Kit (Thermo Scientific, Waltham, MA) with images visualized by an LAS-3000 imaging system (GE Healthcare Biosciences). Post-exposure image processing was limited to linear changes in brightness, contrast, and color balance that were applied to the entire image.

## Statistics

Statistical analyses was conducted using GraphPad Prism software (GraphPad Software, Inc., La Jolla, CA, USA). For (Fig 1A), equations were derived using a linear fit and nonlinear fit (second order polynomial) analysis. For analyses comparing 2 experimental groups (Fig 3A, 3B, and 3C), a Wilcoxon matched pairs signed ranked test was performed. For analysis involving 3 or more experimental groups, a Kruskal-Wallis test post-hoc Dunn's multiple comparisons test was performed. For the CFSE proliferation and RA + IL-6 time course analysis, we performed a two-way ANOVA by comparing the means of the different treatment conditions (i.e., MAdCAM-1 + RA vs MAdCAM-1 + RA + TGF-β).

## Supporting information

**S1 Fig. Inefficient formation of CD69$^+$/CD103$^+$ T$_{RM}$s following MAdCAM-1 costimulation of purified naïve CD4$^+$ T cell cultures.** (A) Representative flow cytometry dot plots of PBMC derived naïve (upper) and bulk (lower) CD4$^+$ T cells costimulated with MAdCAM-1 (+ RA and TGF-β). Y-axis indicates CD69, x-axis indicates CD103. Frequencies of CD69$^+$/CD103$^+$ cell as indicated.
(TIF)

**S2 Fig. VCAM-1 and CD28 mAb exhibit stronger costimulatory signaling and viability compared to MAdCAM-1.**
(A) Representative flow cytometry dot plots and lymphocyte gating of bulk CD4$^+$ T cells that were costimulated with MAdCAM-1, VCAM-1 or CD28 mAb (+ RA and TGF-β). (B) Flow cytometric analysis of the lymphocyte gated population compared between the costimulatory ligands (n = 10). Error bars indicate standard deviation (*: p < 0.05, **: p < 0.01, ***: p < 0.001, ****: p < 0.0001).
(TIF)

**S3 Fig. Naïve cell proliferation and gut surface marker expression following MAdCAM-1 costimulation.** Flow cytometric dot plots from a representative donor naïve CD4$^+$ T cell proliferation by dye dilution (y-axis) and (A) CD103, (B) CD69 and (C) CCR5 expression (x-axis) following MAdCAM-1 costimulation in the presence of RA and TGF-β. Division 0 (d0) is indicated by a black arrow. Division index as indicated.
(TIF)

**S4 Fig. IL-6R expression levels in CD4 + bulk cultures.** (A) CD45RO/CD45RA gating by flow cytometry from a representative donor. (B) IL-6Rα levels of each gated population. B) IL-6Rβ levels of each gated population.
(TIF)

**S5 Fig. Changes in protein expression between D0 Naïve T cells and D5 T cells after treatment with RA + IL-6.**
Measured proteins include (A) α$_4$β$_7$, (B) CCR5 and (C) CCR9. The gates used in Fig 3E are indicated. All plots are from the same representative donor.
(TIF)

**S6 Fig. Schematic of naïve CD4$^+$ T cells priming by MAdCAM-1 in the presence of IL-6, Retinoic Acid and TGF-β.** Schematic representation of the differentiation of naïve CD4 + T cells primed through engagement of α$_4$β$_7$ by MAdCAM-1. Integrin signaling in the presence of Retinoic acid (RA), TGF-β, and cytokine-driven activation of the JAK/STAT pathway promotes tissue-resident memory T cell (T$_{RM}$) differentiation with upregulation of CD69 and CD103 (α$_E$β$_7$).
(TIF)

**S1 Table. List of antibodies used in the study.**
(PDF)

## Acknowledgments

The authors would like to thank Lyle McKinnon and Alexandra Schuetz for discussion and advice. We thank Dr. Robert Yarchoan for providing Ruxolitinib and AZD1480.

## Disclaimer

Material has been reviewed by the Walter Reed Army Institute of Research. There is no objection to its presentation and/or publication. The opinions or assertions contained herein are the private views of the author, and are not to be construed as official, or as reflecting true views of the Department of the Army or the Department of Defense. The investigators have adhered to the policies for protection of human subjects as prescribed in AR 70–25.

## Author contributions

**Conceptualization:** Claudia Cicala.

**Data curation:** Han G. Kim, Amanda Chan, Sinmanus Vimopatranon, Samuel Wertz, Il-Young Hwang, Hana Schmeisser, Madelyn M. Seemiller, Danlan Wei, Livia R. Goes.

**Formal analysis:** Han G. Kim, Amanda Chan, Sinmanus Vimopatranon, Il-Young Hwang, Dawei Huang, James Arthos, Claudia Cicala.

**Investigation:** Alexandre Girard, Andrew Jiang.

**Resources:** John H. Kehrl, Paolo Lusso, Elena Martinelli, James Arthos, Claudia Cicala.

**Software:** Dawei Huang.

**Supervision:** Claudia Cicala.

**Writing – original draft:** James Arthos, Claudia Cicala.

**Writing – review & editing:** Han G. Kim, John H. Kehrl, Paolo Lusso, Marcelo Soares, Elena Martinelli, James Arthos, Claudia Cicala.

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
