## [Decision Letter · Decision Letter 0]

9 Jan 2026

PPATHOGENS-D-25-02438

IL-6 is one of the Key Factors in the Formation of Gut Tissue Resident Memory T Cells from

Naïve T cells

PLOS Pathogens

Dear Dr. Cicala,

Thank you for submitting your revised manuscript to PLOS Pathogens. The manuscript was reviewed by one of the previous reviewers and I also evaluated it. After careful consideration, we feel that it has merit but does not fully meet PLOS Pathogens's publication criteria as it currently stands. Therefore, we invite you to submit a revised version of the manuscript that addresses the points raised during the review process.

I had several comments that should be addressed:

1. Could you test blocking IL-6 and IFN-b together when using supernatants?

2. I would have liked the data on viability to be included in a supplementary figure or table. That way the readers can have a full appreciation of what is going on.

3. Sup. Figure 3 doesn’t discriminate between naive and memory CD4 T cells. It is unclear what is the meaning of CD4dim CD3dim since this is never discussed elsewhere in the manuscript. However, we can appreciated from the flow plot that almost all CD4 T cells seem to express IL-6R.

4. “All study participants received written informed consent.” Provided instead of received?

We look forward to receiving your revised manuscript.

Kind regards,

Guido Silvestri

Academic Editor

PLOS Pathogens

Susan Ross

Section Editor

PLOS Pathogens

Sumita Bhaduri-McIntosh

Editor-in-Chief

PLOS Pathogens

orcid.org/0000-0003-2946-9497

Michael Malim

Editor-in-Chief

PLOS Pathogens

orcid.org/0000-0002-7699-2064

**Journal Requirements:**

At this stage, the following Authors/Authors require contributions: Claudia Cicala, Hans Kim, Amanda Chan, Sinmanus Vimonpatranon, Alexandre Girard, Andrew Jiang, Samuel Wertz, Il-Young Hwang, John Kehrl, Hana Schmeisser, Madelyn Seemiller, Paolo Lusso, Dawei Dawei, Danlan Wei, Livia Goes, Marcelo Soares, Elena Martinelli, and James Arthos. Please ensure that the full contributions of each author are acknowledged in the "Add/Edit/Remove Authors" section of our submission form.

4) We notice that your supplementary Figures are included in the manuscript file. Please remove them and upload them with the file type 'Supporting Information'. Please ensure that each Supporting Information file has a legend listed in the manuscript after the references list.

Potential Copyright Issues:

i) Figure 4A and Graphical Abstract. Please confirm whether you drew the images / clip-art within the figure panels by hand. If you did not draw the images, please provide (a) a link to the source of the images or icons and their license / terms of use; or (b) written permission from the copyright holder to publish the images or icons under our CC BY 4.0 license. Alternatively, you may replace the images with open source alternatives. See these open source resources you may use to replace images / clip-art:

6) In the online submission form, you indicated that All data will be available. All PLOS journals now require all data underlying the findings described in their manuscript to be freely available to other researchers, either

1. In a public repository

2. Within the manuscript itself

3. Uploaded as supplementary information.

7) Please amend your detailed Financial Disclosure statement. This is published with the article. It must therefore be completed in full sentences and contain the exact wording you wish to be published.

**Reviewers' Comments:**

Reviewer's Responses to Questions

**Part I - Summary**

Reviewer #2: In this revised manuscript the authors explore the different factors that can mediate

differentiation of naive CD4T cells into GI TRM phenotype like with the ultimate goal to help

understand what intervention can help restore of increase TRM CD4 T cells in the GI which

is known to help with gut mucosa integrity and lower inflammation.

Their main findings are: In PBMCs of healthy donors, MAdCam-1 +RA prime

naïve CD4+T cells to express GI TRM cell markers and that IL-6 and TGFb promote this

differentiation. This is a well writtten paper with compelling data exposing one of potential many pathways involved in this process.

**Part II – Major Issues: Key Experiments Required for Acceptance**

Reviewer #2: All major issues raised by reviewers were addressed by the authors.

**Part III – Minor Issues: Editorial and Data Presentation Modifications**

Reviewer #2: NA

PLOS authors have the option to publish the peer review history of their article (what does this mean? ). If published, this will include your full peer review and any attached files.

**Do you want your identity to be public for this peer review?** For information about this choice, including consent withdrawal, please see our Privacy Policy .

Reviewer #2: No

**Figure resubmission:**
---

## [Editor Report · Decision Letter 1]

3 Mar 2026

Dear Dr. Cicala,

We are pleased to inform you that your manuscript 'IL-6 is one of the Key Factors in the Formation of Gut Tissue Resident Memory T Cells from

Naïve T cells' has been provisionally accepted for publication in PLOS Pathogens.

Best regards,

Guido Silvestri

Academic Editor

PLOS Pathogens

Susan Ross

Section Editor

PLOS Pathogens

Sumita Bhaduri-McIntosh

Editor-in-Chief

PLOS Pathogens

orcid.org/0000-0003-2946-9497

Michael Malim

Editor-in-Chief

PLOS Pathogens

orcid.org/0000-0002-7699-2064
---

## [Editor Report · Acceptance letter]

Dear Dr. Cicala,

We are delighted to inform you that your manuscript, "IL-6 is one of the Key Factors in the Formation of Gut Tissue Resident Memory T Cells from

Naïve T cells," has been formally accepted for publication in PLOS Pathogens.

Best regards,

Sumita Bhaduri-McIntosh

Editor-in-Chief

PLOS Pathogens

orcid.org/0000-0003-2946-9497

Michael Malim

Editor-in-Chief

PLOS Pathogens

orcid.org/0000-0002-7699-2064